# Reactive Extrusion of Maleic-Anhydride-Grafted Polypropylene by Torque Rheometer and Its Application as Compatibilizer

**DOI:** 10.3390/polym13040495

**Published:** 2021-02-05

**Authors:** Asra Tariq, Nasir M. Ahmad, Muhammad Asad Abbas, M Fayzan Shakir, Zubair Khaliq, Sikandar Rafiq, Zulfiqar Ali, Abdelhamid Elaissari

**Affiliations:** 1Polymer Research Lab., School of Chemical and Materials Engineering, National University of Science and Technology, Islamabad 44000, Pakistan; asra_nse3@scme.nust.edu.pk (A.T.); masad_nse02@scme.nust.edu.pk (M.A.A.); fayzan.shakir@ntu.edu.pk (M.F.S.); 2Department of Polymer Engineering, National Textile University, Faisalabad 37610, Pakistan; zubair.khaliq@ntu.edu.pk; 3Department of Chemical, Polymer and Material Engineering, University of Engineering and Technology, Kala Shah Kaku Campus 54890, Pakistan; sikandar@uet.edu.pk; 4Department of Chemical Engineering, COMSATS University Islamabad, Lahore Campus 45550, Pakistan; zulfiqar.ali@cuilahore.edu.pk; 5Univ Lyon, University Claude Bernard Lyon-1, CNRS, LAGEPP-UMR 5007, F-69622 Lyon, France; abdelhamid.elaissari@univ-lyon1.fr

**Keywords:** functionalization, grafting, polypropylene, reactive extrusion, torque analysis, thermal analysis

## Abstract

This study is based upon the functionalization of polypropylene (PP) by radical polymerization to optimize its properties by influencing its molecular weight. Grafting of PP was done at different concentrations of maleic anhydride (MAH) and benzoyl peroxide (BPO). The effect on viscosity during and after the reaction was studied by torque rheometer and melt flow index. Results showed that a higher concentration of BPO led to excessive side-chain reactions. At a high percentage of grafting, lower molecular weight product was produced, which was analyzed by viscosity change during and after the reaction. Percentage crystallinity increased by grafting due to the shorter chains, which consequently led to an improvement in the chain’s packing. Prepared Maleic anhydride grafted polypropylene (MAH-g-PP) enhanced interactions in PP-PET blends caused a partially homogeneous blend with less voids.

## 1. Introduction

Polypropylene (PP) is the second-largest consumable polymer due to its better mechanical properties, flexibility, transparency, low cost, ease in processability, and high chemical and moisture resistance [1]. However, the major drawbacks of PP are high thermal expansion coefficient, poor bonding properties, and susceptibility to oxidation [2]. The blending of PP with another polymer with improved thermal characteristics may be more favorable to optimize its properties. PP is a highly nonpolar polymer, so it has limited compatibility with polar polymers. A stabilized homogenous mixture can be achieved by generating compatibility among the polymeric blends that will lower the interfacial tension [3]. However, the functionalization of PP will alter its characteristics due to structural changes [4,5,6].

Various researchers have investigated the grafting of maleic anhydride (MAH) on polymer chains to make it compatible with polar polymers [4,5,6,7,8,9,10,11,12,13,14,15]. MAH imparts carbonyl functional groups on the backbone of PP and makes it compatible with polymers [7,16]. The solution process was initially used for the grafting of MAH on PP, and high reaction yields were achieved [7,8,17]. However, the solution process has limited applications due to the involvement of solvent. MAH was grafted on PP by melt technique by reactive extrusion at varying MAH concentrations, type and amount of initiator, and processing conditions [4,6,7,11,18,19]. In the grafting of MAH on PP chains at high-temperature, peroxide forms free radicals on the PP chain, thus withdrawing hydrogen atoms while imparting radicals on the PP chain. The MAH ring attaches to the PP chain bearing a radical on it; thus, the grafting process continues until termination [6]. Benzoyl peroxide (BPO) has been widely used as initiator to start the reaction for MAH functionalization on polymer chains and was considered suitable for bulk polymerization reactions [20]. Control of the reaction is very important to avoid excessive chain scission and gel formations, which eventually alter the rheological characteristics of PP. Previously, work has been successfully carried out on grafted MAH on HDPE by different peroxides in the presence of electron donor additives (dimethyl sulfoxide, dimethylacetamide, tri(nonylphenyl) phosphate) using torque of reaction. The formation of gel or crosslinking was controlled by optimizing the peroxide amount and its type [21]. The effect of MAH and dicumyl peroxide (DCP) as initiators on chain scission has also been reported [22]. Chain scission was initially high, which was observed by a decrease in the mixing torque value that eventually stabilizes as the reaction proceeds. The addition of monomers during MAH grafting on PP chains has also been investigated [23,24,25,26]. The grafting of MAH on PP was conducted by adding styrene as co-monomer, and a high grafting yield was obtained at styrene to MAH 1:1 proportion. Due to an equal number of monomer ratio being employed, no side reaction occurred [26]. During the melt grafting of MAH on PP, homo-polymerization of MAH by the attack of the peroxide initiator did not appear at 180 °C to 190 °C temperature [12].

Despite extensive research, the effect of MAH grafting on the structure of PP during and after complete reaction is yet not clear, and this area continues to attract research attention [4]. Furthermore, the occurrence and effect of side reactions during MAH grafting on PP have not been investigated in detail. In consideration of the above, it would be vital from the perspective of both product and process development, to explore in detail the effect of important parameters such as MAH and initiator concentration on the extent of grafting and rheology during the course of reactive extrusion.

The effect of MAH and BPO concentrations on the polymers chains’ behavior throughout the reaction during the reactive extrusion process is discussed in this study. In addition, after-grafting changes in the physical properties of PP are not clear in the literature. Hence, the effect of grafting MAH on PP crystallinity, melting temperature, and melt flow index is evaluated. Apart from torque rheometer, Fourier-transform infrared spectroscopy (FTIR), melt flow index (MFI) and differential scanning calorimetry (DSC) techniques were utilized in this study. To check the effect of Maleic anhydride grafted polypropylene MAH-g-PP as a compatibilizer, grafted PP was added in polyethylene terephthalate (PET) blends at varying concentrations. The effect of this compatibilizer on morphology was analyzed by SEM. Dynamic mechanical analysis (DMA) was done to analyze the thermomechanical behavior of compatibilized blends.

## 2. Experimental

### 2.1. Materials and Methods

MAH (>99% pure) with a density of 1.314 g/cm^3^ and BPO (99% pure) were purchased from Sigma-Aldrich St. Louis, MO, USA. Isotactic PP (Mn ≅ 123.7 Kg/mol) was purchased from LCY Chemicals Corp, Kaohsiung, Taiwan. BPO was used as initiator for the grafting of PP with MAH. The density of PP was 0.908 g/cm^3^. The melt flow index (MFI) was 3.297 g/10 min at 190 °C. Commercial grade acetone was used as solvent for MAH and BPO. The solvents and materials were used without further purification. Film grade PET was provided by Gatron Industries limited Karachi, Pakistan.

### 2.2. Chemical Reaction and Reactive Extrusion Process

Table 1 indicates the number of samples with varying concentrations of MAH and BPO in parts per hundred (phr). The samples were named as PM1~PM5 (varying concentration of MAH) and PB1~PB5 (varying concentration of BPO). MAH and BPO were dissolved in acetone and stirred for 10 min at room temperature. PP pellets were added, and the mixture was placed for at least 3 days at room temperature to allow evaporation of acetone. BPO and MAH adhered homogenously onto the surface of PP pellets after the evaporation of acetone. HAAKE Rheomix OS Lab internal mixer (Thermofisher, Dreieich, Germany) was utilized for free radical reaction of PP and MAH using BPO. The functionalization of PP was carried out by a reactive extrusion process in an internal engineering mixer system.

The barrel of the mixer was preheated at 160 °C until the temperature was stabilized. 40 g of MAH- and BPO-coated PP pellets were added in two equal parts by weight. The extrusion process was carried out at 160 °C with a screw speed of 60 rpm for 10 min. The change in torque was recorded for at least 10 min.

### 2.3. Mechanical Blending of PET and PP with MAH-g-PP

Blends of MAH-g-PP and pure PP with PET were prepared by varying composition in an internal mixer by melt blending. PET was added first then PP and MAH-g-PP were added. Blending was done at 270 °C, 70 rpm for 10 min. Compositions details are detailed in Table 2. Films of PET and PP blends were fabricated by the compression molding machine at 200 °C temperature and 2000 psi pressure.

### 2.4. Process and Physical Characteristics

HAAKE Rheomix lab internal mixer was used at a defined speed (shear rate) and time, and PP flow behavior was recorded as torque value. Rotors’ rpm was 60 at 160 °C and a 10-min reaction was conducted. If viscosity increases inside the reaction chamber, the system gains more energy to maintain the speed of rotors, which generates a signal recorded by a transducer. The value of torque was obtained from the attached transducer of the internal mixer. The torque throughout the reaction was continuously monitored, and variation in values was studied. FTIR was done on PP grafted samples. The spectrum was recorded by a Bruker instrument (Fremont, CA, USA) Model Alpha. PP grafted samples were thermally characterized in a Perkin Elmer DSC 4000 (Waltham, MA, USA), by heating 5–8 mg of the sample at 10 °C/min under a nitrogen (N_2_) atmosphere from ambient temperature to 200 °C. The MFI of PP grafted samples was measured in accordance with ASTM 1238 under the weight of 2.16 kg at 190 °C temperature in a Noselab ATS Plastometer (Nova Milanese MB, Italy). The analysis was carried out 3 times for each reactive extruded sample, and the average value was calculated. Scanning electron microscopy (SEM) model Joel JSM 6490A (Peabody, MA, USA) was done to analyze the blends’ morphology. Dynamic mechanical analysis (DMA) by TA Instruments, New Castle, DE, USA of all the prepared blends in contrast of pure PP and PET was performed using ASTM E1640-13 in bending mode with a dual cantilever. Dual cantilever bending mode was used because of the fragile nature of samples. The sample was run under nitrogen atmosphere at 5 °C/min and 1 Hz frequency.

## 3. Results and Discussion

### 3.1. Reactive Extrusion Process

When the mixing process started, the variation in temperature and torque from the experiment was recorded to analyze the effect of the free radical polymer reaction on the melt viscosity of PP. Figure 1a,b describes the torque variation with reaction time by varying BPO and MAH content, respectively.

The recorded instantaneous torque (ᴛ) is correlated with the viscosity (η) of the material in the reaction at temperature *T* for time *t* according to the below-mentioned Equations (1)–(3) [27].
(1)$$\eta =\;\frac{\tau }{\gamma }\;$$
where:
η= dynamic viscosity (Pa·s)τ= shear stress (N/cm^2^)γ= shear rate (sec-1)
(2)$$\tau =\;\frac{ᴛ}{2\pi R_{s}^{2}L}$$(3)$$\gamma =\;\frac{2\omega R_{c}^{2}R_{s}^{2}}{X^{2}\left(R_{c}^{2}-\;R_{s}^{2}\right)}$$
where:
ᴛ= torque (Nm)L= effective spindle length (m)$R_{s}=\;$spindle radius (m)$R_{c}=\;$container radius (m)ω= rotational speed (radians/s)X= radial location

During the reaction time, all quantities are constant except torque. Hence,
(4)$${ᴛ}_{\left(t,T\right)}\;\propto \;\eta_{\left(t,\;T\right)}$$

In all samples, initial high filling peaks appear due to the addition of solid PP (MAH and BPO coated) pellets into the mixer. Torque on the curve shows continuous variation. This unbroken disturbance reflects the feeding and molten accumulation of PP. A high value of torque is visible during the initial 2–4 min due to friction, high viscosity, and surface melting of PP pellets [28].

It is observed that during the initial few minutes, torque highly fluctuates because of radical reaction on PP chains. This results in an increase in viscosity due to a number of radicals formation and MAH molecules. With the passage of time, the fusion of material takes place along with chain breakage and decrease in the molecular weight by grafting of the MAH functional group on PP [2,3]. The molecular weight of PP is possibly decreased due to chain scission in the free radical polymerization process [2]. By reduction in molecular weight, the viscosity decreased. Hence, the possible effects produce a change in torque. The experimental observations indicate that the value of torque stabilizes at the end of the reaction after all the material seems to completely melt. Equilibrium is achieved between shear heating and constant chamber temperature, which results in stable torque value [28]. The increase in the amount of MAH or BPO reduces the time required to reach the steady-state torque value. However, the effect on reaction time by varying the MAH concentration is more visible as shown in Figure 1b. High BPO concentration (Figure 1a) and increasing MAH percentage at constant BPO (Figure 1b) led towards chain scission and crosslinking reaction. The change in torque value remained highly visible (Figure 1b) when the amount of MAH is increased, which shows that high concentration of MAH leads towards more grafting reaction and chain scission [6]. However, high loadings of BPO led to excessive side-chain reactions such as crosslinking [21]. The stabilized value of torque in both the cases at the ending of the reaction time indicates that the chemical reaction did not move towards excessive crosslinking or complete degradation of PP [28]. The stable torque value was higher than the initial torque value due to slight crosslinking in chains.

A possible reaction mechanism [22] during the grafting of MAH on PP by using BPO as initiator is given in Figure 2. The radical is produced either as phenyl radical or benzoyl radical. These radicals attack on the main chain of carbon in polypropylene and extract an hydrogen atom, leaving a radical on the PP chain. On this produced active site, the ring of maleic anhydride is attached.

Crosslinking density in PP by adding MAH and BPO was calculated using the difference between initial and final torque values [21]. An increase in the value of crosslinked density shows that more complex structures are formed as well as gel formation. The gel formation is not required and reduces the grafting percentage. Figure 3 explains the trends in crosslinking density by varying MAH and BPO concentration. It is clearly visible that when BPO amount is increased at constant MAH, overall, the crosslinking density increases due to a large number of free radicals that attack on the PP chain and produce many complex structures and gel structures. However, at constant BPO, initially at 1.0 phr MAH, crosslinking density increases due to side chain reactions; however, after this concentration, free radicals possessed more available sites for attack, and the reaction moved towards grafting [21].

### 3.2. Carbonyl Index of Grafted PP

FTIR spectra of pure polypropylene (PP) and highest MAH-grafted PP (PM4) are shown in Figure 4. In both spectra, peaks are observed at 2950 cm^−1^ due to the asymmetric stretching in the methyl group (-CH_3_). At 2915 cm^−1^ peak is due the asymmetric stretching in the methylene group (-CH_2_-), 2870 cm^−1^ for -CH_3_ symmetric stretching, and 2840 cm^−1^ for -CH_2_-symmetric stretching. Bending peaks of -CH_2_- and –CH_3_ are observed at 1455 cm^−1^ and 1370 cm^−1^, respectively. In all grafted samples, two peaks appeared at 1750 cm^−1^ for the carbonyl (C=O) group of the five-membered ring anhydride and the C=C peak at 1655 cm^−1^. From the FTIR spectra of all PP grafted samples, the carbonyl index (CI) was calculated using Equation (5) [5].
(5)CI= A1750A1455
where $A_{1750}$ is the area of absorbance peak at 1750 cm^−1^, that is characteristic peak of the carbonyl functional group from five-membered cyclic anhydrides; $A_{1455}$ is the area of absorbance peak at 1455 cm^−1^, which is characteristic of the CH_2_ and is proportional to the concentration of PP. Table 3 displays the CI values for the prepared samples. A higher CI value of MAH-grafted PP indicates higher grafting [5].

The value of CI is directly proportional to the percentage of grafting; hence, the sample that shows a high carbonyl index is exhibiting high grafting percentage. Table 3 shows that PM4 has a high CI value and hence high % grafting.

### 3.3. Crystallinity and Melting Temperature of Grafted Samples

The values of heat of fusion obtained by DSC measurements for pure PP and grafted PP samples were utilized to calculate the percentage crystallinity by Equation (6) [6].
(6)$$\%\;\mathrm{Crystallinity}=\;\frac{∆H_{f}^{*}}{∆H_{f}^{100}}\;$$
where $∆H_{f}^{*}$ is the heat of fusion of grafted PP and $∆H_{f}^{100}$ is the fusion heat for a theoretical 100% crystalline PP [6]. 207 J/g is the value used for the enthalpy of fusion for 100% crystalline PP obtained from literature [29]. A comparison of the DSC thermograms of processed samples by reactive extrusion with varying MAH content is shown in Figure 5.

All grafted samples showed a high $∆H_{f}$. On the base of $∆H_{f}$, the calculated % crystallinity is shown against different MAH and BPO content for grafted PP in Figure 6.

It is observed that in all processed samples, the percentage of crystallinity is much higher. This behavior is due to the addition of BPO and MAH, which causes degradation of the PP chains into shorter chains [5]. Chain scission caused a reduction in molecular weight and further reduced entanglement in chains. This reduction enabled a rise in the degree of order of PP chains and hence caused an increase in overall crystallinity [4,5]. When BPO content increased, chains scission increased due to a high level of degradation; thus, the percentage crystallinity remained high for all samples [5]. However, after increasing the MAH concentration at 1.5 phr, percentage crystallinity is noted to be reduced. The reason for this slight fall is MAH attachment on chains, which reduced chain packing [30].

Figure 7 shows the variation in melting temperature ($T_{m}$) by altering MAH and BPO amounts. The structural changes are caused by the addition of MAH on PP chains, which slightly influenced $T_{m}$. Variation in MAH at constant BPO and in BPO at constant MAH first initiated the reduction of $T_{m}$ in processed samples, followed by a sudden rise.

The fall in T_m_ at low concentrations of MAH and BPO is caused by the chains breakage and formation of lower molecular weight chains. However, at higher concentrations of MAH and BPO, complex molecular structures started to appear; thus, $T_{m}$ increased [4,5].

### 3.4. Melt Flow Index (MFI)

MFI values for the functionalized PP samples are given in Table 4.

Figure 8 displays the variation in MFI by altering MAH and BPO amounts. When adding low content of MAH and BPO, the MFI value increases remarkably. By adding MAH and BPO, chains scission occurs due to termination by disproportionation and chain transfer, and due to the presence of possible side reactions. It can be inferred that a combination reaction for termination is less probable than chain transfer or disproportionation [5]. Shorter chains with low molecular weight caused a high flow rate. As the concentration of MAH and BPO was further increased, complex molecules started to form, which resulted in lower MFI values [5].

### 3.5. Morphology by Scanning Electron Microscopy (SEM)

Figure 9 and Figure 10 display the SEM images of pure polyethylene terephthalate (PET), pure polypropylene (PP) and compatibilized and un-compatibilized PP/PET blends, respectively.

Figure 9 shows that pure PP and PET possess totally uniform microstructures and that there are no phase boundaries. However, the blends of PP and PET showed phase boundaries. The presence of MAH-g-PP in PET and PP blends promoted the formation of very fine and dispersed morphology, better adhesion, and partial uniformity (Figure 10a) than an un-compatibilized blend (Figure 10b). This increase in fineness of the prepared blends showed lesser voids and hence decreased the passing of small molecules across the films and overall reduced its water permeability. The blend having no MAH-g-PP showed phase separation and voids due to lack of compatibility between PET molecules and PP chains [31,32,33]. Figure 11 shows the comparison of PET and PP blends by changing the concentration of MAH-g-PP.

In Figure 11, two blends having different compatibilizer concentration are compared, and the SEM image of 5% MAH-g-PP shows less voids that are possibly owing to the presence of uneven physical interactions between the PET and PP chains. In the blends of 2.5% and 5% MAH-g-PP, agglomerates started to appear, which shows the immiscibility between PP and PET molecules in some regions of the blends. PET molecules began to adhere with same type of molecules because of its higher possibility of interactions among the similar kind of molecules to overall stabilize the system [34].

Fracture analysis of all the prepared blends were carried out by taking SEM images on the edges of films shown in Figure 12. The spherical shaped beads present in the figure show the PP material, and PET is the main matrix material. In uncompatibilized blend, it can be seen that the spherical beads are becoming debonded from the main matrix that is PET and this is because of the lack of adhesion on interface. Many holes are visible in un-compatibilized samples due to the drawing of weakly adhered molecules and less physical interactions. Morphology of compatibilized blend of PP and PET blends showed smaller bead size as compare to uncompatibilized blend owing to the greater physical interactions present. Spherical shaped beads of PP polymer now appeared to be attached to the matrix material which is PET by developing bridges. The appeared physical interactions are due to dipole-dipole attraction forces among the carbonyl group of PET and the maleic anhydride group grafted on PP chains present as compatibilizer. There are plane surface and fibrils extensions on the fractured side of compatibilized blends that show the compatibilized PP/PET blend is moderately ductile material [32]. An optimum concentration of MAH-g-PP used as compatibilizer provided partial homogeneous blend of PP/PET with lesser phase separations.

### 3.6. Dynamic Mechanical Analysis

Figure 13 shows the temperature dependence on the tan delta of PP/PET compatibilized blends as compare to pure PP and pure PET.

A sharp prominent peak appeared at around 85 °C in the graph of pure PET that represents the T_g_ of pure PET and it is nearly the same as the data obtained from the analysis of DSC. Two small shoulders are visible in the graph of pure PP at −15 °C and 70 °C. At −15 °C, which is the T_g_ of isotactic polypropylene, β transitions occur in PP. At higher temperature, 70 °C, α transition occurred, which is related to the PP crystalline fracture [35]. However pure PET showed larger area under the peak in comparison with pure PP curve; hence, PET has a higher ability to dissipate energy on application of load in contrast with pure PP. The chains of PP displayed elasticity in material structure, which indicates more load storage ability than dissipation. The partial miscibility of both the components in PET/PP blends is evident as the peak started to merge. In all blends, the peak at low-temperature is started to shift near the high-temperature peak. This behavior is because of the presence of MAH-g-PP in PET and PP blends, which allowed physical interaction between the components and the partially homogenous blends’ formation. The area under the peak also enhanced in all prepared blends as compared to pure PP, which revealed high impact bearing properties of blends [36,37,38]. Figure 14 clarifies the effect of temperature on the value of storage modulus of prepared compatibilized blends in contrast to pure PP and pure PET.

In pure PET, a lowest value of storage modulus can be seen and which was noted to be regularly decreased by increasing temperature. It was observed that within this temperature range, PET did not show melting but only a slight softening by increasing temperature, while crystal melting was observed in long-range temperature. Furthermore, PP exhibited a higher value of storage modulus as compare to pure PET; however, a sudden decrease in the value of storage modulus of PP at 0 °C appeared proceeded with a high variation in G′ due to β transition in PP. By adding compatibilizer in the PET/PP blend, the value of storage modulus rose in contrast to both pure PP and PET. This increase in storage modulus indicates a rise in the stiffness of the polymer due to a restriction in the segmental motion. This raise indicates a high physical interaction and improved compatibility between PP and PET chains. By increasing the concentration of MAH-g-PP, the storage was not pronounced owing to the presence of uneven physical interaction, which eventually led to improper PET/PP phase adhesion. The presence of MAH-g-PP decreased the transition region in the blend of PP and PET. It is worth to notice that with the rise in temperature, the prepared samples appeared to obtain a small difference in the storage modulus value, and curves started to come close. With increasing temperature, polymer chains started to move, and softness in samples appeared. It is thus concluded that the compatibilizer increased the value of storage modulus of the blend, which approaches to high stiffness in samples due to the annealing of the films at room temperature [36,37].

Figure 15 presents the change in loss modulus by increasing temperature for the compatibilized blends of PET/PP in contrast to pure PP and pure PET. It displays the energy dissipation for prepared samples. The value of loss modulus and storage reduces due to the smaller force needed for deforming the sample. Initially, all samples hinder the segmental motion of molecules, but with a rise in temperature, the molecular motion of this type is activated. From figure the T_g_ of PET is 82 °C that is nearly the same value found from DSC analysis. The T_g_ of pure PP appeared at −15 °C. By forming a compatibilized blend of PET and PP, in CB1, the T_g_ of PP slightly shifted to about 10 °C, and the T_g_ of PET was unnoticeable. It can be inferred that CB1 has improved compatibility between PP and PET components. CB2 and CB3 also presented an increase in the value of T_g_ of PP; and these two compatibilized blends also displayed the T_g_ of PET at 90 °C temperature [36,37].

## 4. Conclusions

In this study, grafting of PP by MAH was carried out using a torque rheometer. Variations in PP structure during the reaction and after grafting were studied by torque evolution and flow behavior. It was found that the highest percentage of grafting was achieved at 0.2 phr MAH and 0.4 phr BPO, since increasing the amount of MAH and BPO from the said values started side-chain reactions and crosslinking. However, at high grafting percentage, the molecular weight decreased, and lower viscosity at high flow rate was observed. This decrease in viscosity is due to chain scission in the free radical polymerization reaction. The high amount of BPO favors more side-chain reactions, which is why the amount of BPO should be controlled to less than 0.4 phr. Chains breakage caused an increase in percentage crystallinity, which was found by heat of fusion of MAH-g-PP samples. Grafting on PP chains also showed a slight change in melting temperature (1 °C to 3 °C) analyzed by DSC thermograms owing to chains breakage. The study showed that free radical polymerization yielded a high grafting percentage at the expense of molecular weight. Side reactions occurred that caused structural changes that eventually effected the flow behavior of PP. MAH-g-PP provided excellent compatibilization for synthesizing homogeneous PET and PP 60/40 ratio. However, with an increase in the amount of MAH-g-PP greater than 1% in 60/40 PET and PP ratio, agglomeration started to appear, reducing the compatibility between the phases.

## Figures and Tables

**Figure 1 polymers-13-00495-f001:**
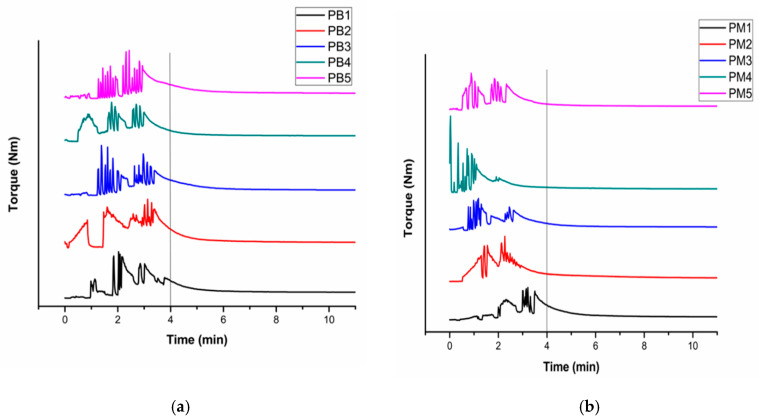
Torque variation of PP with reaction time by altering (**a**) BPO concentration and (**b**) MAH concentration.

**Figure 2 polymers-13-00495-f002:**
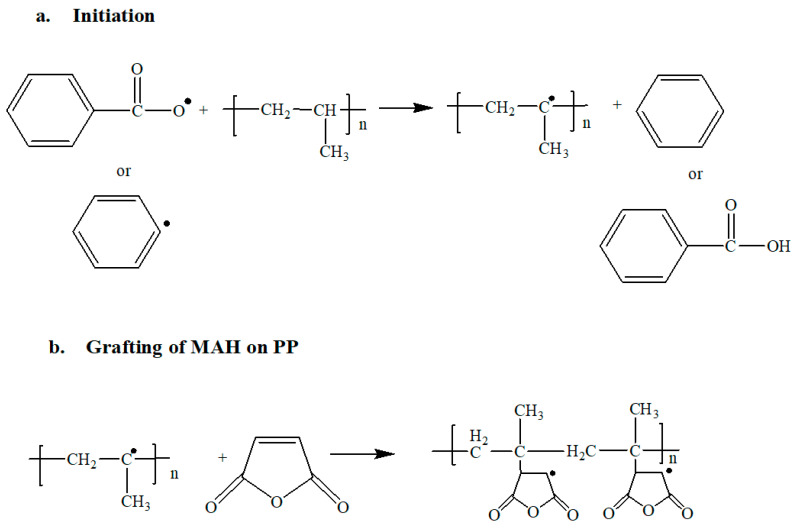
(**a**) Initiation reaction for the grafting by free radical polymerization. (**b**) Grafting of MAH ring on the backbone of PP.

**Figure 3 polymers-13-00495-f003:**
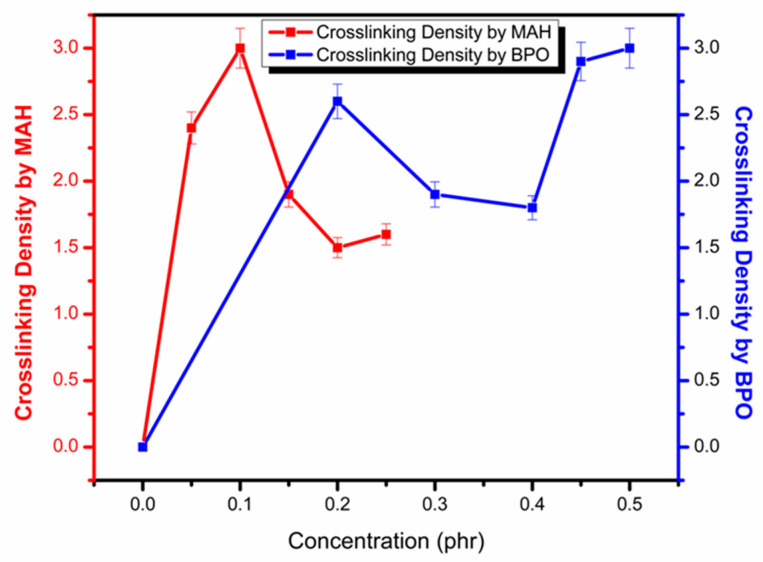
Effect of MAH and BPO concentration on the crosslinking density during the reaction.

**Figure 4 polymers-13-00495-f004:**
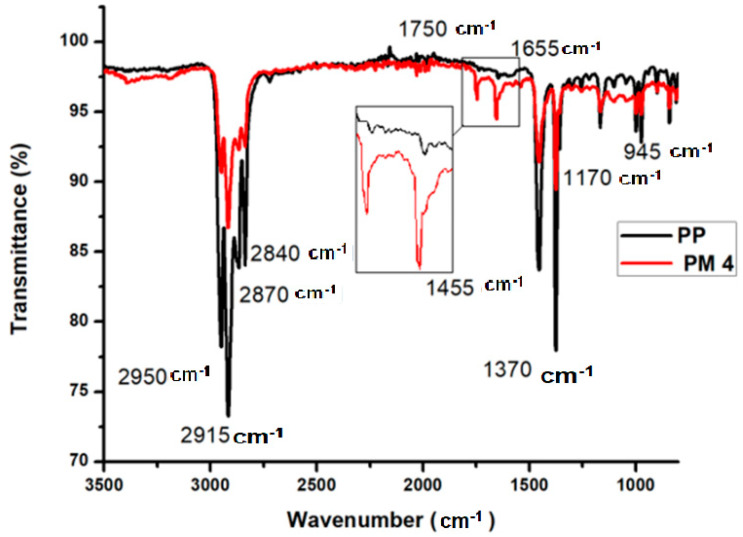
Comparison of the spectra of pure polypropylene (PP) and highest maleic anhydride (MAH) grafted PP obtained by Fourier Transform Infrared Spectroscopy (FTIR).

**Figure 5 polymers-13-00495-f005:**
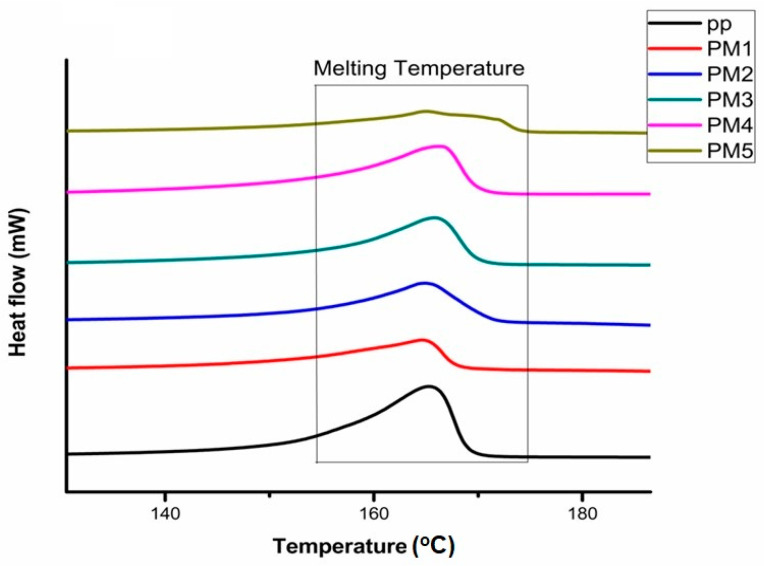
Differential scanning calorimetry (DSC) thermograms for MAH-grafted PP samples for varying MAH contents.

**Figure 6 polymers-13-00495-f006:**
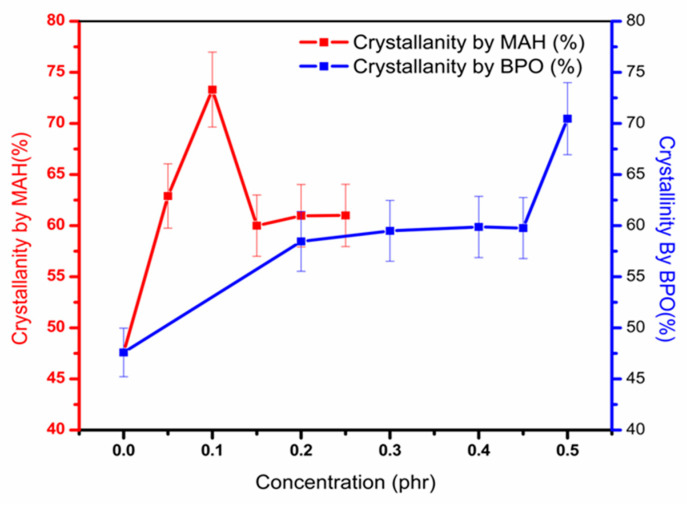
Effect on percentage crystallinity for grafted samples by varying MAH and BPO content.

**Figure 7 polymers-13-00495-f007:**
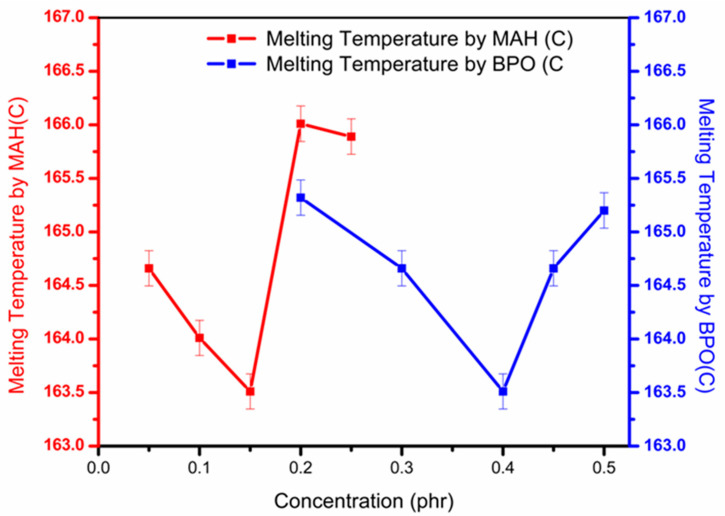
Effect on melting temperature (T_m_) for grafted samples by varying MAH and BPO contents.

**Figure 8 polymers-13-00495-f008:**
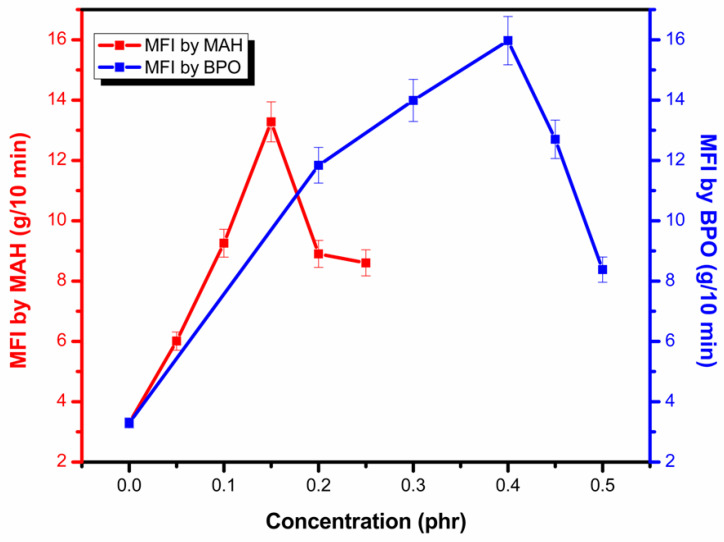
Melt flow index of functionalized samples by varying MAH and BPO.

**Figure 9 polymers-13-00495-f009:**
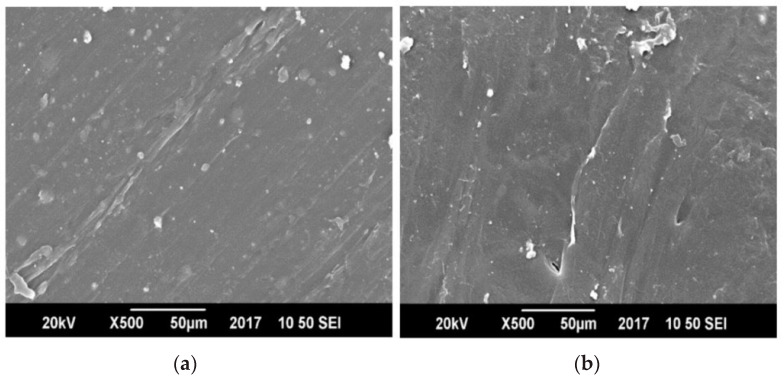
Scanning Electron Microscopy (SEM) images of (**a**) pure polyethylene Terephthalate (PET) and (**b**) pure polypropylene (PP).

**Figure 10 polymers-13-00495-f010:**
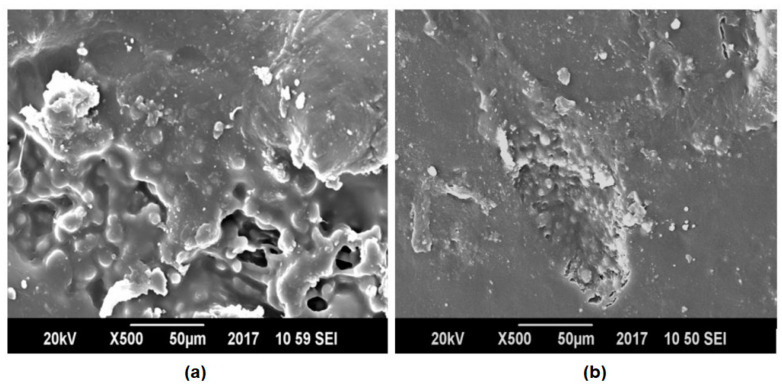
Scanning Electron Microscopy (SEM) images of (**a**) PP/PET and (**b**) Maleic Anhydride grafted Polypropylene/ Polypropylene/ Polyethylene Terephthalate (MAH-g-PP/PP/PET) blends with composition 40/60 and 2.5/37.5/60, respectively.

**Figure 11 polymers-13-00495-f011:**
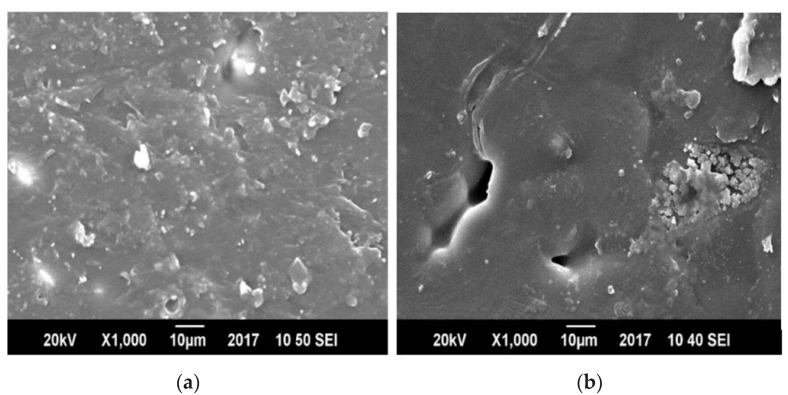
Scanning Electron Microscopy (SEM) images of (**a**) MAH-g-PP/PP/PET and (**b**) MAH-g-PP/PP/PET blends with composition 2.5/37.5/60 and 5/35/60, respectively.

**Figure 12 polymers-13-00495-f012:**
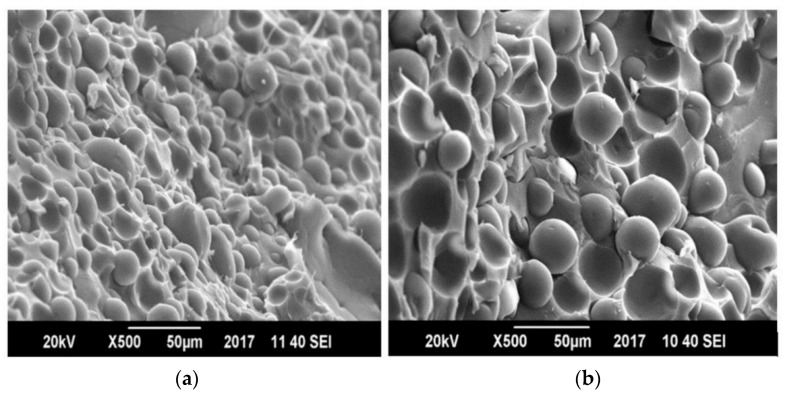
Scanning Electron Microscopy (SEM) images of fracture analysis of 60% PET (**a**) 5% MAH-g-PP/PET blends and (**b**) un-compatibilized PP/PET.

**Figure 13 polymers-13-00495-f013:**
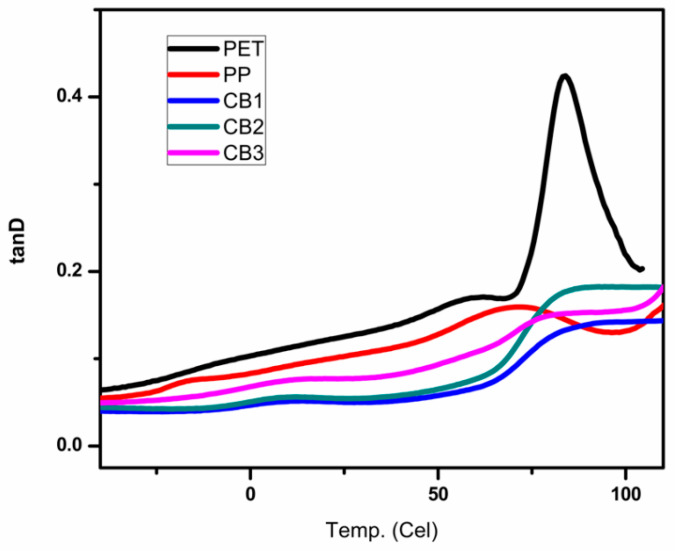
Variations in the tan delta by increasing temperature for prepared samples analyzed by Dynamic Mechanical Analyzer (DMA) dual cantilever in bending mode.

**Figure 14 polymers-13-00495-f014:**
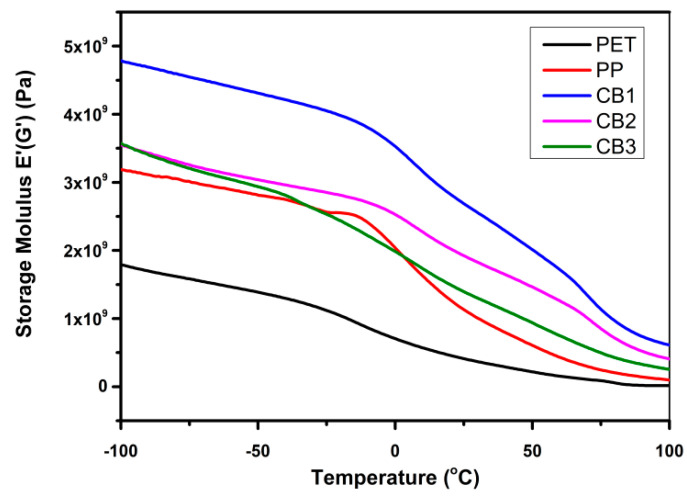
Variations in storage modulus by increasing temperature for prepared samples measured by DMA dual cantilever in bending mode.

**Figure 15 polymers-13-00495-f015:**
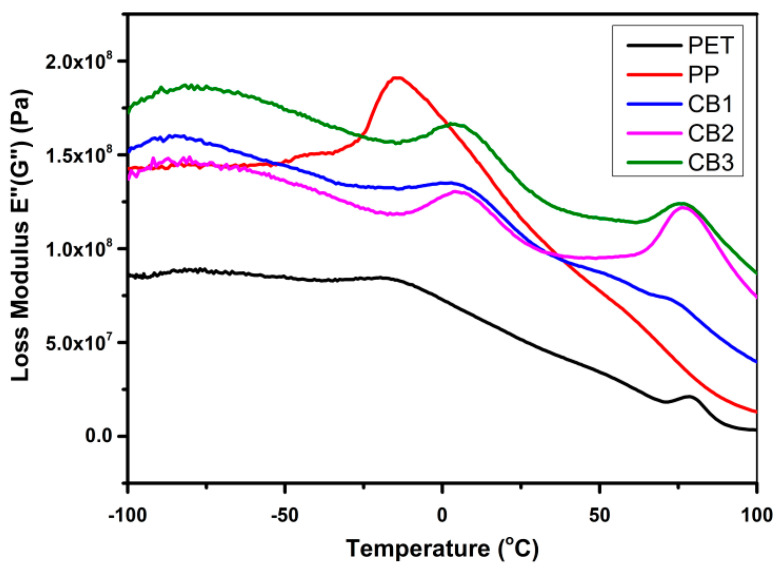
Variations in the value of loss modulus with increasing temperature for prepared samples measured by DMA dual cantilever in bending mode.

**Table 1 polymers-13-00495-t001:** Experimental design by varying Maleic Anhydride (MAH) at constant Benzoyl Peroxide (BPO), and varying BPO at constant MAH.

Sample Name	MAH (phr)	BPO (phr)	Sample Name	MAH (phr)	BPO (phr)
PM1	0.05	0.4	PB1	0.15	0.2
PM2	0.10	0.4	PB2	0.15	0.3
PM3	0.15	0.4	PB3	0.15	0.4
PM4	0.20	0.4	PB4	0.15	0.45
PM5	0.25	0.4	PB5	0.15	0.5

**Table 2 polymers-13-00495-t002:** Composition details of Polypropylene (PP) /Polyethylene terephthalate (PET) PP/PET blends.

Samples	Film Grade PET (%)	Isotactic PP (%)	MAH-g-PP (%)
CB 1	60%	39% PP	1%
CB 2	60%	37.5% PP	2.5%
CB 3	60%	35% PP	5%
CB 4	60%	40% PP	-
CB 5	100%	-	-
CB 6	-	100%	-

**Table 3 polymers-13-00495-t003:** Carbonyl Index (CI) values of all grafted samples.

Sample Name	CI Value	Standard Deviation (±)	Sample Name	CI Value	Standard Deviation (±)
PM1	0.24	0.012	PB1	0.25	0.0125
PM2	0.37	0.0185	PB2	0.27	0.0135
PM3	0.38	0.019	PB3	0.38	0.019
PM4	0.41	0.0205	PB4	0.36	0.018
PM5	0.40	0.02	PB5	0.36	0.018

**Table 4 polymers-13-00495-t004:** Melt flow index for processed samples and pure PP.

Sample Name	MFI (g/10 min) at 2.16 kg and 190 °C	Standard Deviation (±)	Sample Name	MFI (g/10 min) at 2.16 kg and 190 °C	Standard Deviation (±)
PP	3.297	0.16485			
PM1	6.011	0.30055	PB1	11.84	0.592
PM2	9.256	0.4628	PB2	13.99	0.6995
PM3	14.28	0.664	PB3	15.97	0.7985
PM4	8.901	0.44505	PB4	12.70	0.635
PM5	8.604	0.4302	PB5	8.380	0.419

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
