# Peer review of "Reactive Extrusion of Maleic-Anhydride-Grafted Polypropylene by Torque Rheometer and Its Application as Compatibilizer"

_polymers, 2021, doi:10.3390/polym13040495_

Round 1

Reviewer 1 Report

The summary indicates the following (line 21-23): “This study is based upon the functionalization of Polypropylene (PP) by radical polymerization to optimize its properties by influencing the molecular weight”. But you do not appreciate the bleaching or novelty of knowing and studying the different parameters, this punctual, it is advisable to novelty this stop increases attention.

The introduction is appropriate, presents a comprehensive analysis of the state of the art, identifying the area of opportunity in the line of research. In this sense they mention that the main drawbacks of PP are the high coefficient of thermal expansion, the poor bonding properties and susceptible to oxidation. In this sense they identify the different studies on the subject and clarify that it is vital to know the effect of grafts on the structure of PP during the reaction and after the complete reaction, since it is not yet clear and this area continues to attract research efforts. The novelty and contribution of the study is defined.

If presented appropriately.         

The experimental methodology is properly described.

 (Lines 133-137): Expand and clarify differences with respect to variation, especially in the initial times where a difference is observed for each system. Figure 1b and 1b, could go in a single line, to compare and discuss in more detail.

(Lines 162-166): No es claro explicación respecto a reacciones propuestas (Fig 2).  Calidad de imágenes puede ser mejorada.

(Line 170): “The molecular weight of PP is possibly decreased”  Why?

Explanation in Figure 3, it is recommended to improve with scientific support. Focusing on why you look at it.

The discussion of results is presented in general, further analysis is recommended.

Figures 9 and 10, has different size and supposedly is the same scale. Be careful when handling images. The decrease in size should be respected without affecting the scale. So it can be comparative. Same scale/size for all micrographs.

(Lines 267-272): Where the wording is noted. Improve discussion.

Conclusions can be improved. Since they look like observations and not conclusions of the way they are presented. It should focus on why each of the results occurred.

Author Response

  1. (Lines 133-137): Expand and clarify differences with respect to variation, especially in the initial times where a difference is observed for each system. Figure 1b and 1b, could go in a single line, to compare and discuss in more detail.

Reply. Figure and 1a and 1b is added in single line for better comparison. In all samples, initially there is high disturbance in the value of torque and it shows a continuous variation in the value because initially the material is added in the mixer and pellets just started to melt and radical reaction was started. As the value of torque is related to the force required to move the inside material so initially in all samples there is fluctuations due to friction, high viscosity, and surface melting of PP pellets. However, the time required to achieve equilibrium varied by varying MAH and BPO concentration and the increase in the amount of MAH or BPO reduced the time required to reach steady-state torque value. However, the effect on reaction time by varying MAH concentration is more visible as shown in Fig. 1b. High BPO concentration (Fig. 1a) and increasing MAH percentage at constant BPO (Fig. 1b) led towards chains scission and crosslinking reaction. The change in torque value remained highly visible (Fig. 1b) when the amount of MAH is increased this shows that high concentration of MAH leads towards more grafting reaction and chain scission. However, high loadings of BPO, led to excessive side-chain reactions such as crosslinking.

The mentioned figures are now more clearly explained by rearranging some data and adding more detail in manuscript, the explanation for these figures are now starting from page#6, line 150-168 in revised version.

  1. (Lines 162-166): No es claro explicación respecto a reacciones propuestas (Fig 2).  Calidad de imágenes puede ser mejorada.

Reply. This figure is redrawn and it shows the possible reaction mechanism during the grafting of MAH on PP by using BPO as initiator. The radical is produced either as phenyl radical or benzoyl radical. These radicals attacked on the main chain of carbon in polypropylene and extracted hydrogen atom by leaving radical on PP chain. On this produced active cite, ring of maleic anhydride attached.

The added data highlighted on page# 7, line 176-178 in revised version

  1. (Line 170): “The molecular weight of PP is possibly decreased”  Why?

Reply. The molecular weight of PP is possibly decreased due to chain scission in free radical polymerization process. During chain scission, polymer chains are randomly break from various point and over all weight of polymer is reduced.

  1. Explanation in Figure 3, it is recommended to improve with scientific support. Focusing on why you look at it.

The discussion of results is presented in general, further analysis is recommended.

Reply. Figure 3 is now explained in more detail, Crosslinking density in PP by adding MAH and BPO was calculated using the difference in initial and final torque value. Increase in the value of crosslinked density shows that more complex structures are formed and gel formation. The gel formation is not required and it reduces the grafting percentage. Fig. 3 explains the trends in crosslinking density by varying MAH and BPO concentration. It is clearly visible that when BPO amount is increased at constant MAH, overall, the crosslinking density increased due to a large number of free radicals that attacked on PP chain and produced many complex structures and formation of gel structures. However, at constant BPO, initially at 1.0 phr MAH, crosslinking density increased due to side chain reactions, however, after this concentration, free radicals possessed more available sites for attack and reaction moved towards grafting

The details are highlighted on page#7, line 179-187 in revised version.

  1. Figures 9 and 10, has different size and supposedly is the same scale. Be careful when handling images. The decrease in size should be respected without affecting the scale. So it can be comparative. Same scale/size for all micrographs.

Reply. Thank you. This is correct now

  1. (Lines 267-272): Where the wording is noted. Improve discussion.

Reply.   In sem images, prepared blends were compared with pure PP and PET. The blends are also compared to check the effect of the prepared compatibilizer on the compatibility of two phases in blends. As reference the images of blend without compatibilizer is used. The details are more comprehensive now on page#12-13, line#263-278 in revised version.

  1. Conclusions can be improved. Since they look like observations and not conclusions of the way they are presented. It should focus on why each of the results occurred.

Reply. In this study, grafting of PP by MAH was carried out by the torque rheometer. Variations in PP structure during the reaction and after grafting was studied by torque evolution and flow behavior. It was found that highest percentage of grafting was achieved at 0.2 phr MAH and 0.4 phr BPO since increasing the amount of MAH and BPO from said value started side-chain reactions and crosslinking. However, at high grafting percentage, molecular weight decreased and lower viscosity at high flow rate was observed. This decrease in viscosity is due to chain scission in free radical polymerization reaction. The high amount of BPO favors more side-chain reactions and that’s why the amount of BPO should be controlled to less than 0.4 phr. Chains breakage caused an increase in percentage crystallinity that was found by heat of fusion of MAH-g-PP samples. Grafting on PP chains also showed a slight change in melting temperature (1oC to 3oC) analyzed by DSC thermograms owing to chains breakage. The study showed that free radical polymerization gave high grafting percentage at the expense of molecular weight. Side reactions occurred that caused structural changing which eventually effected the flow behavior of PP. MAH-g-PP provided excellent compatibilization for synthesizing homogeneous PET and PP 60/40 ratio. However, with the increase in the amount of MAH-g-PP in blend higher than 1% in 60/40 PET and PP ratio, agglomeration started to appear that reduces the compatibility between phases.

Reviewer 2 Report

This manuscript described the preparation of maleic anhydride-grafted polypropylene by reactive extrusion for using as compatibilizer for pp/pet composite. The results are somewhat interesting and can be considered for publication.

line 59, the full name for DCP initiators should be given.

line 133, "---MAH and BPO content, respectively" may be "--BPO and MAH content, respectively" because fig. 1a showed BPO results and fig 1b showed MAH results, respectively.

line 167, figure 2a should be redrawed. For example, "phenyl radical" may be benzoyl oxide radicals, noted as phCOO*, ph is the benzenic rings, and the graft chains should be rewritten as a polymer chain.

line 176-177, check MAH and fig.1a and fig.1b

Author Response

  1. line 59, the full name for DCP initiators should be given.

Reply. Thank you for mentioning. The full name of DCP is added and highlighted on page#2, line 56 in revised version.

  1. line 133, "---MAH and BPO content, respectively" may be "--BPO and MAH content, respectively" because fig. 1a showed BPO results and fig 1b showed MAH results, respectively.

Reply. Thank you for correction. The mentioned change is done as advised on page#4, line 128.

  1. line 167, figure 2a should be redrawed. For example, "phenyl radical" may be benzoyl oxide radicals, noted as phCOO*, ph is the benzenic rings, and the graft chains should be rewritten as a polymer chain.

Reply. Figure 2a is redrew as recommended and replaced with corrected. There is the possibility that both phenyl or benzoyl radical may generate that can further reaction with PP chains, extract hydrogen atom and start the grafting of maleic anhydride on PP chains. Correction is done on page#6, line 167

  1. line 176-177, check MAH and fig.1a and fig.1b

Reply. Recommended error is corrected and highlighted on page#6,line 163-167

Round 2

Reviewer 1 Report

The different questions issued in prior evaluation were answered. So the quality and analysis of results improved. Conclusions can be improved.

Reviewer 2 Report

The authors have well reviewed this manuscript, and the present form can be accepted for publication.